# Statistical Measurements and Club Effects of High-Quality Development in Chinese Manufacturing

**DOI:** 10.3390/ijerph192316228

**Published:** 2022-12-04

**Authors:** Chunyan Lin, Wen Qiao

**Affiliations:** School of Statistics and Mathematics, Shandong University of Finance and Economics, Jinan 250014, China

**Keywords:** high-quality development of manufacturing, club convergence, nonlinear time-varying factor model, ordered logit model

## Abstract

Advanced manufacturing is the pillar for building a modern economic system. We measured the level of high-quality development of manufacturing (HQDM) in China, and found that it has gone through the three stages of expansion, cultivation, and promotion. Spatially, it is characterized as “high in the east, low in the west” and “fast in the west, slow in the east”, and presents non-equilibrium characteristics. To overcome the subjective bias introduced by artificially set clubs, we utilize a data-driven nonlinear time-varying factor model for clustering into four convergent clubs, where provinces with higher intensity of environmental regulation and environmental preference tend to move closer to the clubs with a higher level of HQDM. We reveal the convergence patterns and regional differences in HQDM, which provides a new perspective for determining the trends of high-quality manufacturing development, thus allowing for policy recommendations targeted at narrowing the manufacturing development gap.

## 1. Introduction

The new era of technological revolution and industrial change is still emerging, and the main contradiction of Chinese society now is the contradiction between the people’s growing need for a better life and the unbalanced and insufficient mode of development. In order to cope with the new changes in technology and meet the new needs of the people, the Chinese economy must follow the path of high-quality development. Manufacturing is the lifeblood of the Chinese economy and is crucial for enhancing the country’s strategic competitiveness, and the high-quality development of manufacturing (HQDM) is an important strategic task for Chinese economic development in the new development stage. China has the world’s largest and most mature manufacturing system [1]. According to the World Bank, Chinese manufacturing value added has continued to grow (Figure 1), and has remained in first place in the world for 12 consecutive years since 2010, with its position as a major manufacturing country becoming increasingly secure. However, there are still quality problems, such as over-capacity and environmental pollution [2], and regional disparities have caused widespread concern. Figure 1 shows that, for more than a decade, the growth rate of manufacturing value added has been slower than that of GDP, and the proportion of GDP (p) has presented a decreasing trend. As the development of industrialization has entered a higher stage of development, it is necessary to continuously promote the transformation and upgrading of manufacturing and to improve its quality and efficiency. Therefore, in order to complete the leap “from big to strong” and ensure the strong manufacturing nature of the country, we must unswervingly promote the HQDM. Due to the significant differences among Chinese provinces, in terms of geographic location, level of economic development, and policy orientation, there exists a significant regional imbalance in the HQDM, which affects the overall national development level. Therefore, it is of great theoretical and practical significance to systematically study the convergence characteristics of Chinese provincial HQDM levels and relevant influencing factors.

Convergence studies provide an important way to analyze regional disparities and convergence characteristics. They have not only been deeply applied in economics fields, such as economic growth [3] and Internet finance [4], but also widely promoted in green innovation [5], human capital [6], carbon emissions [7], green total factor productivity [8], energy productivity [9], and so on. Convergence theory originated from neoclassical economic growth theory, and its classical meaning refers to the process of catching up and convergence of economic development levels of less-developed economies to developed economies in the long run, including σ convergence, β convergence, and club convergence. In particular, σ convergence is the gradual reduction of dispersion in different economies over time, while β convergence is divided into conditional convergence and absolute convergence. The former argues that different economies have different initial conditions and structural characteristics, and that convergence can only occur between economies with similar structural characteristics. The latter indicates that different economies converge to the same steady-state in the long run, regardless of the differences in initial conditions and structural characteristics between economies. The concept of “convergence club” was first introduced by Baumol (1986), who argued that countries (or regions) with similar initial endowments, geographic locations, and cultural systems have similar degrees of economic convergence, and that different countries (or regions) have different clubs in development, based on their initial conditions. The convergence hypothesis has been studied empirically in a large number of countries and regions; however, due to the complexity and diversity of the real world, the hypothesis has not yet been universally confirmed, which has attracted more and more economists to conduct more in-depth research.

The topic of manufacturing convergence and its influencing factors has been discussed from the following two aspects. First, in terms of research on manufacturing convergence, academics have looked at output [10], technology [11], labor productivity [12], energy productivity [13], CO_2_ emissions [14], and eco-efficiency [15] for quantitative analysis of manufacturing convergence, and the results of most studies support the existence of convergence characteristics in different aspects of manufacturing. Specifically regarding the convergence of the Chinese manufacturing industry, the identification methods have mostly considered σ or β convergence. The σ convergence is analyzed mainly by the coefficient of variation method, while the β convergence test mainly draws on the convergence theory proposed by Barro and Sala-i-Martin, in order to construct a convergence model for analysis, on the basis of which a spatial convergence model is constructed considering spatial effects. For example, Liu D. (2022) studied the convergence of manufacturing energy carbon efficiency using coefficient of variation and convergence models, and found that manufacturing energy carbon efficiency is not σ convergent, but has β convergence [16]. Yang C. et al. (2021) observed strong conditional β convergence characteristics of low-carbon innovation in the Chinese manufacturing industry through a conditional convergence model, and examined the effect of economic openness on β convergence by incorporating spatial spillover effects into the convergence function [17]. Erban A. C. et al. (2022) examined the β convergence in high-technology manufacturing sectors in the EU28 countries, they found that the new EU member states displayed a higher β convergence rate than the EU15 countries did [18]. However, both σ and β convergence testing methods are somewhat biased [19]. For this reason, some studies have used the club convergence method. For example, Huang et al. (2018) considered three groups of clubs when testing the club convergence of energy efficiency—namely, eastern versus mid-western cities, resource-based versus non-resource-based cities, and environmentally focused versus non-environmentally focused cities—which were grouped based on similarities in economic indicators or regimes [20]. This artificial division lacks objective and reasonable criteria [21], and may hide potential club convergence [22]. Second, research has been conducted on the factors influencing convergence. Studies have been conducted to examine the influencing factors or convergence mechanisms of club convergence using ordered logit models or probit models regarding issues such as per capita income [23], renewable energy technology innovation [24], R&D expenditure [25], and energy intensity [22]; however, most studies on the convergence of the Chinese manufacturing industry have only sought to determine whether it exists or not [5,16], and few have explored the factors influencing the convergence of HQDM, lacking an examination of the convergence mechanisms.

Based on the above, in this paper, we attempt to expand and enrich the existing research in the following ways: First, methods based on artificially predetermined clubs can hardly meet the prerequisite of “similar initial level and structural characteristics,” as proposed in the concept of club convergence, which will reduce the scientificity of club identification, to a certain extent. To address this issue, we use a nonlinear, time-varying factor model, a data-driven clustering approach, and convergence analysis to construct the study object. This endogenous identification method takes into account the time-varying heterogeneity among individuals, can compensate for the shortcomings of existing studies, and can avoid the bias caused by manual classification [24]. Meanwhile, it has been demonstrated in the literature that such a model is applicable for analysis of the Chinese case [26]. Therefore, we introduce this method to study the convergence of Chinese HQDM, in order to automatically screen and identify convergence clubs of Chinese HQDM. Second, as mentioned previously, there is a lack of research on the influencing factors of manufacturing club convergence taking Chinese provinces as samples. In this paper, we use an ordered logit model to explore the causes of club convergence of Chinese HQDM, then test the convergence mechanism of regional HQDM (see Figure 2).

## 2. Materials and Methods

### 2.1. Two-Stage Entropy Method

In this paper, the two-stage entropy method is used to measure the level of HQDM in each province of China, using the algorithm for layer-by-layer empowerment and multidimensional weighting. The entropy method is generally based on macroscopic data and, compared with subjective evaluation methods, it is not influenced by subjective judgment and has a certain degree of scientific objectivity [27]. The principle of the entropy method is to measure index weights based on the information entropy of the original data: the smaller the entropy value, the greater the weight given, indicating that the index is more important. To date, the entropy method has been widely used in manufacturing [2,28], economics [17], innovation capability [5], and environmental pollution [1] fields, among others. The calculation formula is detailed in the following.

First, in order to achieve comparable indicators and accurate index measurement, the raw data were standardized.

Positive indicators:(1)$$z_{ijk}=\frac{x_{ijk}-\min\left\{x_{1jk},\cdots ,x_{mjk}\right\}}{\max\left\{x_{1jk},\cdots ,x_{mjk}\right\}-\min\left\{x_{1jk},\cdots ,x_{mjk}\right\}}$$

Negative indicators:(2)$$z_{ijk}{}^{\prime }=\frac{\max\left\{x_{1jk},\cdots ,x_{mjk}\right\}-x_{ijk}}{\max\left\{x_{1jk},\cdots ,x_{mjk}\right\}-\min\left\{x_{1jk},\cdots ,x_{mjk}\right\}}$$
where i indexes the provinces (i=1,2,⋯,m), j denotes the measurement indicators of HQDM (j=1,2,⋯,n), k denotes the measurement dimension (k=1,2,⋯,h), $x_{ijk}$ denotes the raw data for indicator j of province i within dimension k, $\max\left\{x_{1jk},\cdots ,x_{mjk}\right\}$ is the maximum value of the indicator in all years, $\min\left\{x_{1jk},\cdots ,x_{mjk}\right\}$ is the smallest value of the indicator in all years, and $z_{ijk}$ and $z_{ijk}{}^{\prime }$ are the standardization results.

We calculated the weight of indicator j of province i in dimension k as follows:(3)$$w_{ijk}=\frac{z_{ijk}}{\sum_{i=1}^{m}z_{ijk}}$$
and calculated the entropy value of indicator j of dimension k as
(4)$$e_{jk}=-\frac{1}{\ln m}\sum_{i=1}^{m}w_{ijk}\ln w_{ijk}$$

We calculated the weights of the indicators of dimension k of province i as
(5)$$g_{ik}=\frac{\sum_{j=1}^{n}\frac{1-e_{jk}}{\sum_{j=1}^{n}e_{jk}}w_{ijk}}{\sum_{i=1}^{m}\sum_{j=1}^{n}\frac{1-e_{jk}}{\sum_{j=1}^{n}e_{jk}}w_{ijk}}$$
the weighted entropy value of dimension k as
(6)$$e_{k}=-\frac{1}{\ln m}\sum_{i=1}^{m}g_{ik}\ln g_{ik}$$
and the HQDM index for province i as
(7)$$I=\sum_{k=1}^{h}\frac{1-e_{k}}{\sum_{k=1}^{h}\left(1-e_{k}\right)}g_{ik}$$

The larger the HQDM index I, the higher the level of manufacturing development in the province.

### 2.2. Club Convergence Test Method

We took the level of HQDM in 30 Chinese provinces from 2008 to 2019 as an empirical basis, and endogenously identified the club convergence of HQDM in Chinese regions based on the nonlinear, time-varying factor model proposed by Phillips and Sul (2007) and their clustering algorithm [29]. Unlike the traditional artificial grouping, this method can avoid a priori sample separation and imposes no special requirements regarding the smoothness characteristics of the data [30]. Specifically, the identification method for club convergence consists of three components: convergence testing of nonlinear, time-varying factor models, club clustering, and club integration.

#### 2.2.1. Convergence Tests of Nonlinear, Time-Varying Factor Models

We decomposed the level of HQDM $I_{it}$ in year *t* for province i as:(8)$$I_{it}=\delta_{i}\mu_{t}+\varepsilon_{it}$$
where $\mu_{t}$ is the public factor, $\varepsilon_{it}$ is a random perturbation term, and $\delta_{i}$ reflects the heterogeneous distance between $\mu_{t}$ and $I_{it}$. Although $\delta_{i}$ reflects individual heterogeneity, this heterogeneity does not change over time and, so, the random perturbation term $\varepsilon_{it}$ needs to be incorporated into the coefficients to obtain the nonlinear, time-varying factor model.
(9)$$I_{it}=\left(\frac{\delta_{i}\mu_{t}+\varepsilon_{it}}{\mu_{t}}\right)\mu_{t}=\delta_{it}\mu_{t}$$
where $\delta_{i}$ represents the change in individual heterogeneity over time, including the random perturbation term $\varepsilon_{it}$. Thus, Equation (9) contains the time-varying characteristics of individual heterogeneity.

In order to model the time-varying parameters, a relative transfer coefficient was defined:(10)$$h_{it}=\frac{I_{it}}{\frac{1}{N}\sum_{i=1}^{N}I_{it}}=\frac{\delta_{it}}{\frac{1}{N}\sum_{i=1}^{N}\delta_{it}}$$
(11)$$H_{t}=\frac{1}{N}\sum_{i=1}^{N}{\left(h_{it}-1\right)}^{2}$$
where $h_{it}$ reflects the degree of dispersion of the level of HQDM in province i from the average level of each province and its trend over time, and $H_{t}$ is the cross-sectional variance of $h_{it}$ in year t. As the common growth path (i.e., the common factor) is partially eliminated, when there is convergence, $h_{it}\to 1$ and $H_{t}\to 0$ are set.

Further, in order to test the original hypothesis of convergence, it was necessary to construct a semiparametric model:(12)$$\delta_{it}=\delta_{i}+\frac{\sigma_{i}\xi_{it}}{L\left(t\right)t^{\alpha }}$$
where $\delta_{i}$ is a constant term related only to the qualities of the level of HQDM in province i and does not change over time; $\sigma_{i}$ is a scale parameter for heterogeneity; $\xi_{it}\sim \mathrm{iid}(0,1)$, which is weakly correlated with t; L(t) is a slowly varying function, and when t→∞, L(t)→∞; α is the decay rate, where a larger α yields faster convergence. This semiparametric model indicates that convergence holds as long as α≥0, then $\delta_{it}\to \delta_{i}$.

Based on the above derivation process, the original and alternative hypotheses of the convergence test can be written as:$$H_{0}:\delta_{i}=\delta \;\&\;\alpha \geq 0\;H_{1}:\delta_{i}\neq \delta \;\mathrm{or}\;\alpha <0$$

To test the original hypothesis of convergence, regression was performed using the following equation:(13)$$\log\left(\frac{H_{1}}{H_{t}}\right)-2\log L\left(t\right)=\hat{a}+\hat{b}\log t+{\hat{u}}_{t}$$
where L(t)=logt,t=[γT],[γT]+1⋯T, γ>0, and γ is a parameter that determines the starting time t. Since T=12<50 in this paper, γ=0.3. Furthermore, $\hat{a}$ is the estimated value of α, and $\hat{b}$ is fitting coefficient of logt, where $\hat{a}=0.5\hat{b}$. The HAC one-sided *t*-test was used to test the original hypothesis of α≥0. If t<−1.65, it means that the original hypothesis of convergence can be rejected at the 5% level of significance. The above test is called the logt test.

#### 2.2.2. Club Clustering

The logt test is the basic test premise to determine convergence. To obtain convergence clubs, Phillips and Sul developed a data-driven clustering algorithm, using the following steps [29].

Sorting

The mean values of HQDM level for each province in the final period are ranked in order from highest to lowest, and the value of the time span parameter f is taken as 1/3. The sorting is based on:(14)$$\left(T-{\left[Ta\right]}^{-1}\right)\sum_{t=\left[Ta\right]+1}^{T}I_{it},a=1-f$$

2.Select core group

Among the sorted panel data, the k provinces with the highest means of $I_{it}$ are used as the basis, while other provinces are added in turn as alternative groups of clubs $G_{k}\left(2\leq k\leq N\right)$. A logt test is subsequently performed for each alternative group, and the statistics $t_{k}=t\left(G_{k}\right)$ are calculated. The core group $G_{k^{*}}$ containing $k^{*}$ provinces is filtered according to the following criteria:(15)$$k^{*}=argmax_{k}\left\{t_{k}\right\},s.t.\min\left\{t_{k}\right\}>-1.65$$

One province is added to the alternative core group at a time, and values of $t_{k}$ within the group are calculated until the end, when $t_{k}<-1.65$ (i.e., $t_{k}$ is initially less than the critical value at the 5% significance level). The first k members form a total of k−1 core alternative groups, and the largest value of $t_{k}$ among them is selected. The corresponding alternative group becomes the core club $G_{k^{*}}$. If $t_{k}<-1.65$ for the first two provinces, the first province is removed and the above steps are repeated for the remaining provinces until $t_{k}>-1.65$ is satisfied. If $t_{k}>-1.65$ does not occur, it is judged that the provinces and cities are divergent, and no convergence club $G_{k^{*}}$ is formed.

3.Add provinces.

Adding provinces to the core group in turn and performing logt tests, the t-statistic obtained from this regression is expressed as $\hat{t}$. If $\hat{t}>c$ (c is the set threshold, c=0), the province is retained in the core club $G_{k^{*}}$. Similarly, the remaining provinces are added to the test following the above steps, and the core group $G_{k^{*}}$, together with the newly added provinces, form the first convergence club. Then, a logt test is performed on the first converging club, in order to ensure that the entire group satisfies $t_{\hat{b}}>-1.65$. If $t_{\hat{b}}\leq -1.65$, the critical value c is raised to improve the judgment of the logt test, and this step is repeated until $t_{\hat{b}}>-1.65$ for the first converging club.

4.Stop algorithm

Continue the logt test for all provinces that did not enter the first convergence club. If $t_{\hat{b}}>-1.65$, the remaining provinces mentioned above form the second club. If $t_{\hat{b}}\leq -1.65$, steps 1–3 are repeated for the remaining provinces above, in order to determine whether a smaller convergence club exists.

#### 2.2.3. Club Integration

As the convergence clubs formed by clustering after the above steps were obtained by increasing the critical value c, different clubs still have the possibility of convergence at the 5% significance level, and it is necessary to merge the clubs satisfying $t_{\hat{b}}>-1.65$. Drawing on the study of Schnurbus et al., the clubs were tested for merging through the following steps: assume that there are a total of M converging clubs, Club1,Club2,⋯ClubM, and that a logt test is conducted for each two adjacent clubs in turn, yielding a total of $(M-1)t_{m}$ values. If $t_{m}>-1.65$ and $t_{m}>t_{m+1}$, the two clubs can be merged into one club; otherwise, the two clubs are retained. If $t_{m}>-1.65$ for the last two groups, they are combined into one club.

### 2.3. Data Source and Processing

The sample data were mainly obtained from China Statistical Yearbook, China Science and Technology Statistical Yearbook, China Industrial Statistical Yearbook, China Environmental Statistical Yearbook, China Energy Statistical Yearbook, China High Technology Industry Statistical Yearbook, China Labor Statistical Yearbook, provincial and municipal statistical yearbooks, CSMAR, EPS, China Economic Network and World Brand Lab, and so on. A total of 30 provincial-level administrative regions were studied in this paper, while Tibet was excluded due to serious missing data. Brand value data were calculated by summing the brand values of each brand, according to provinces, given in the analysis report of China’s 500 Most Valuable Brands from 2008–2019. In addition, in order to maintain the integrity of the sample, individual missing data were filled in by interpolation and analogy.

## 3. Results

### 3.1. Measurement of HQDM in China

#### 3.1.1. Constructing the Index System

We combined the connotation of HQDM and constructed a HQDM evaluation index system containing five primary indicators and 26 secondary indicators, including innovation-driven, quality-first, green development, structural optimization, and talent-based indicators (Table 1).

Innovation-driven.

Innovation is the core driver of social development and the key motivating force to achieve high-quality development. Only an innovation-driven economy can achieve sustainable high-quality development, and we should adhere to the innovation-driven development path and drive innovation output through innovation input. Innovation input is measured by secondary indices $X_{1}-X_{4}$, and innovation output is measured by secondary indices $X_{5}-X_{9}$.

2.Quality-first.

Quality is the lifeline of building a strong manufacturing country. We must pay attention to product quality, improve production efficiency, and take the road of winning by quality. The production efficiency of manufacturing industry was mainly reflected in terms of labor productivity ($X_{10}$), and the ultimate purpose of technological introduction ($X_{11}$) and equipment transformation ($X_{12}$) is also to improve product quality and optimize production, indirectly reflecting the quality level. The brand value level ($X_{13}$) can reflect the quality brand building from the side; in addition, the “quality first” policy of manufacturing can also be reflected by $X_{14}-X_{15}$.

3.Green development.

Green sustainability is an important focus for building a strong manufacturing country, being a key element of high-quality development. One of the main lines of China’s economic policy since the 18th Party Congress is to adhere to green development, reduce pollutant emissions, and promote the implementation of the goals of carbon peaking and carbon neutrality. The environmental secondary indicators, composed of $X_{16}-X_{19}$, reflect the main inputs and results of cities in managing the environment.

4.Structural optimization.

Structural adjustment is a key part of building a strong manufacturing country. In addition to strengthening the transformation and upgrading of traditional manufacturing industries and promoting industrial integration, we must also cultivate high-end industries, optimize the industrial structure, and take the development path of improving quality and increasing efficiency. Structural optimization includes both structural integration and structural upgrading. Structural integration is reflected by the secondary indicators $X_{20}-X_{21}$, while structural upgrading is measured by $X_{22}-X_{23}$.

5.Talent-based.

Talent is fundamental in constructing a manufacturing powerhouse. The accumulation of human capital is the root cause of sustained economic growth [31]. Only by taking talents as the basis and following the development path led by talents can we better promote the HQDM. To be talent-based, it is not only necessary to grow the talent team, but also to improve the treatment of talents, attract high-end talents, and establish a sound scientific and reasonable employment mechanism. Therefore, this dimension can be measured by indicators such as $X_{24}-X_{26}$.

#### 3.1.2. Measurement Results

Using the two-stage entropy method introduced earlier for step-by-step calculation, the HQDM indices of 30 Chinese provinces from 2008–2019 were obtained (Table 2), and the HQDM levels of each province were analyzed according to the calculation results. Table 3 shows the measurement results for the HQDM level of each province from 2008 to 2019, and Figure 1 reflects the changes in the national and regional HQDM indices from 2008 to 2019.

Similar to the results of existing studies [32], we observed an overall growth trend in the HQDM index from 2008 to 2019, indicating an improvement in the quality of manufacturing development nationwide. According to the annual growth rate of the development index, China’s HQDM shows obvious stage characteristics, roughly divided into three stages: expansion, cultivation, and promotion.

The first stage is the expansion period (2008–2014), where China’s HQDM level rose rapidly, with the HQDM index increasing from 0.231 to 0.352, with an average annual growth rate of 7.27%. Since its accession to the WTO in 2001, China’s manufacturing industry has been deeply integrated into the global industrial chain with its unique comparative advantages, and its status as the “world’s factory” has been increasingly consolidated, especially since the outbreak of the global financial crisis in 2008, which provided a favorable opportunity for China’s manufacturing industry to adjust its layout and optimize its structure. Manufacturing value added grew rapidly, from USD 1.48 trillion to USD 3.48 trillion, over the same period, indicating that the significant expansion of manufacturing scale led to a significant increase in HQDM levels.

The second stage is the cultivation period (2015–2017). As China’s economic development enters the new normal, the Chinese government has actively promoted the transformation of the manufacturing industry from the crude scale and speed type to the intensive quality and efficiency type, such as the State Council issuing Made in China 2025, establishing the National Leading Group for the Construction of a Strong Manufacturing Country, strengthening the HQDM coordination planning, and other measures to strongly promote the transformation and upgrading of the manufacturing industry. During this period, China’s HQDM level showed obvious characteristics of speed shift, as the HQDM index only increased from 0.354 to 0.368, with the average annual growth rate slipping to 1.49%.

The third stage is the promotion period (2018 to present). With the new expression of high-quality development, proposed for the first time at the 19th Party Congress in 2017, the Party Central Committee and the State Council have attached more importance to the implementation of high-quality economic and social development, and HQDM has been further strengthened. The effects of a series of policies introduced in the previous period to support the cultivation of new dynamic energy in the manufacturing industry and the structural reform on the supply side have gradually emerged, and the average annual growth rate of HQDM index rebounded to 5.68%.

At the regional level (Figure 3), the level of HQDM in China showed a significant regional imbalance. First of all, the level of HQDM in China presented a “high in the east, low in the west” trend; that is, the overall development level of the eastern coastal provinces was significantly higher than that of the central and western regions. On one hand, this may be due to the superior geographical location, developed transportation facilities, and more open trade development in the eastern region, where local governments pay high attention to environmental pollution control, such as taking the lead in formulating environmental control policies and investing large amounts of human, material, and financial resources [32,33,34]. On the other hand, the higher level of industrial agglomeration in the eastern region has continued to generate an increase in economic productivity [35], which has led to an overall increase in the level of HQDM. Secondly, from the viewpoint of the average annual growth rate, the initial level of the western provinces was relatively low, and the growth rate of HQDM was faster, with the characteristic of “slow in the east, fast in the west”. The average annual growth rate of the manufacturing development index in the west was 6.26%, which was 1.14% and 1.31% higher than that in the east and the center, respectively, indicating a “catch-up effect” that has not previously been mentioned in the existing literature [32,33,34]. On one hand, this may be due to the low base of the western region; on the other hand, it may also be a “latecomer advantage”, as the Chinese government has attempted to replicate the successful experience of HQDM in the eastern region, including institutional policies, advanced technology, and central financial support, in the central and western regions, thus allowing the level of HQDM in the central and western regions to improve relatively quickly. Therefore, there may be a convergence trend of HQDM in China. In the following, this issue is discussed, based on the nonlinear, time-varying factor model and its clustering algorithm proposed by Phillips and Sul (2007).

### 3.2. Club Convergence Analysis of HQDM in China

#### 3.2.1. National and Traditional Club Area Convergence Results

Before performing club convergence identification, we first performed a club convergence test on the level of HQDM in 30 provinces across the country, in order to determine whether club convergence is present in the country as a whole. The test results (Table 3) indicated that the national $\hat{b}$ value corresponded to a t value of −11.0874, less than the critical value of −1.65 at the 5% level of significance, thus rejecting the original hypothesis of overall convergence. This indicates that the national HQDM level does not show an overall convergence trend under the premise of considering heterogeneity and time-varying speed of convergence. Although there was no overall convergence in the HQDM levels of the 30 provinces nationwide, this does not exclude the possibility of club convergence and, so, further club convergence tests are required.

We first applied the traditional artificially predetermined club method to divide the 30 provinces and cities into three major economic regions—namely, east, central, and west (the eastern region includes Beijing, Tianjin, Hebei, Liaoning, Shanghai, Jiangsu, Zhejiang, Fujian, Shandong, Guangdong, and Hainan; the central region includes Shanxi, Jilin, Heilongjiang, Anhui, Jiangxi, Henan, Hubei, and Hunan; the western region includes Inner Mongolia, Guangxi, Chongqing, Sichuan, Guizhou, Yunnan, Tibet, Shaanxi, Gansu, Qinghai, Ningxia, and Xinjiang)—based on traditional geographic information division criteria, and tested for club convergence in the three major economic regions. The test results (Table 3) indicated that the t values corresponding to the eastern, central, and western $\hat{b}$ values were −11.9733, −11.1466, and −4.7495, respectively, which were also all less than −1.65 at the 5% significance level, indicating that there was no regional convergence characteristic of HQDM level within the traditional three major economic regions, also indicating that the traditional method of artificially presetting clubs is not feasible and does not accurately reflect the intrinsic connection and similarity of regional HQDM.

#### 3.2.2. Club Convergence Test and Integration Test

In order to find the initial convergence club, we used the club convergence identification method described previously to analyze the convergence of HQDM levels in each province of China. This method can better overcome the defects caused by artificial division and strong assumptions in the traditional convergence test, in order to identify the convergence characteristics and convergence clubs of Chinese HQDM more scientifically, allowing us to summarize their developmental commonalities and explore the reasons for their differences.

First, we applied the logt test and clustering algorithm as the initial club test for the 30 provinces in China (Table 4). The test results indicated that the t values obtained by fitting the four initial clubs at the 5% significance level were all greater than −1.65, indicating that the original hypothesis of HQDM convergence for each initial club held.

Subsequently, we performed a merger test on the four initial clubs obtained, according to the club integration method, in order to identify whether the original inter-clubs could be merged to form larger convergent clubs. The results of the merger test for adjacent initial clubs showed that the t values obtained from the fits of merged Clubs A and B, merged Clubs B and C, and merged Clubs C and D were less than −1.65 at the 5% significance level, which means that the four clubs could not be integrated to form a larger club. Therefore, the initial clubs did not satisfy the conditions of the club merger test, as shown in the last column of Table 4, thus eventually yielding 4 convergent clubs comprising 2, 2, 16, and 10 provinces, respectively (Table 5).

Table 5 depicts the basic characteristics of the final four convergence clubs. By comparing the map of China, there was no obvious connection between the geographical distribution and the obtained clubs [22]. The mean HQDM values of the four clubs were 0.6035, 0.4763, 0.3315, and 0.2303, respectively, in decreasing order. To further explore the dynamic change characteristics of these four convergence clubs, the relative transfer paths of each convergence club also needed to be examined.

#### 3.2.3. Relative Transfer Path Results for Each Convergence Club

According to Phillips et al. (2009), although club members will converge to the same steady-state over time, there will be different convergence paths for each member, due to the heterogeneity of the initial state. To further explore the HQDM convergence path in each province, we plotted the relative transfer paths of HQDM levels in each convergence club member province for 2008–2019 by calculating the relative transfer coefficient $h_{it}$ for each convergence club member province (see Figure 4, Figure 5, Figure 6 and Figure 7).

From the convergence paths of the member provinces of Club 1, although the initial levels of HQDM in Jiangsu Province and Guangdong Province were similar, the gap between Jiangsu Province and Guangdong Province, in terms of HQDM level, widened during 2010–2014, where the HQDM level of Jiangsu Province was always higher than that of Guangdong Province. After 2015, the level of HQDM of Guangdong Province was in the state of “accelerated catching up”, gradually narrowing the gap with Jiangsu Province, and successfully overtaking Jiangsu Province and moving into the “leading” position in 2017.

From the convergence paths of Club 2 member provinces, Shanghai and Zhejiang HQDM presented the phenomenon of “catching up with each other”, with Shanghai’s HQDM level being higher than Zhejiang during 2009–2011, Zhejiang’s HQDM level catching up with Shanghai during 2012–2015, Shanghai’s HQDM level briefly exceeding Zhejiang in 2016, and Zhejiang catching up with Shanghai again after 2017.

The convergence paths of the member provinces of Club 3 can be roughly divided into high-, medium-, and low-level groups, according to the initial HQDM level. The high-level group included Shandong province, Beijing city, and Tianjin city, whose initial HQDM level was high in 2008 and was in the “leading” state during the observation period of 2008–2019. The medium-level group included Sichuan Province, Hunan Province, Hubei Province, Hebei Province, Henan Province, Anhui Province, and Fujian Province, whose HQDM levels remained around the club average during 2008–2019; however, Anhui Province and Fujian Province succeeded in “catching up” with Tianjin City in the high-level group during 2018–2019. The low-level group included Liaoning, Chongqing, Jiangxi, Shaanxi, Guizhou, and Yunnan provinces, whose initial level of HQDM in 2008 was low and an overall “catching up” status was presented in the period 2008–2019, among which Guizhou and Yunnan provinces had the most obvious “catching up” characteristics, which gradually narrowed the development gap with other club member provinces during 2008–2019.

Looking at the convergence paths of the member provinces of Club 4, the HQDM level of Hainan Province was always in the “leading” position during 2008–2019, while the “catching up” feature was most obvious in Qinghai Province. Other provinces, such as Gansu, Guangxi, and Heilongjiang, generally remained near the average level of the club.

Based on the members included in each convergence club, it can be seen that the member provinces belonging to the same convergence club differed significantly from the traditional geographic division of the situation, and they were entirely endogenously determined based on the HQDM data of each province. Combined with Figure 4, the geographical location of each club member was further examined. The results indicated that, in terms of the geographical distribution of each province, Clubs 1 and 2 both include two eastern regions; Club 3 includes six eastern, five central, and five western regions; Club 4 includes one eastern, three central, and six western regions. It can be seen that there was no obvious similarity in the geographical distribution of the member regions in each club, which also indicates, to some extent, that the division of clubs based only on a single indicator such as geographical location or other criteria lacks rationality, and also demonstrates that the factors influencing the convergence of HQDM clubs in China are more complex and need to be further explored.

### 3.3. Factors Influencing Club Convergence

The four convergence clubs based on HQDM in China were obtained by “letting the data speak for itself,” as detailed above. A question thus arose: what are the factors that determine whether a province belongs to a certain convergence club? In the following, we detail our further investigation of the factors influencing the convergence clubs based on the ordered logit model.

#### 3.3.1. Variable Selection

The following five variables were selected, in order to examine their influence on the convergence clubs based on the HQDM: (1) environmental regulation (ER)—as an effective means of environmental protection by the government, the intensity of environmental regulation affects manufacturing production decisions to a certain extent, and can stimulate technological innovation and promote transformation and upgrading, which may affect the convergence of the HQDM. The frequency of environmental protection words in the government work report was chosen to reflect the intensity of environmental regulation. The higher the word frequency, the greater the intensity of environmental regulation, and vice versa; (2) environmental preference (EP)—we used the urban greening coverage to measure this variable; (3) openness (OPEN)—foreign trade can increase technology spillover, promote industrial upgrading, and contribute to the high-quality development of manufacturing, which we measured in terms of the share of total imports and exports in GDP; (4) organizational management change (OMC)—measured by the share of the number of non-state enterprises in industrial enterprises above the scale; (5) urban population density (UPD)—urban population density can promote manufacturing industrial upgrading by releasing investment and consumption demand, which, in turn, affects HQDM club convergence. Referring to Huang et al. (2018), population per unit area was used to measure this variable [20]. The descriptive statistics of the variables are provided in Table 6.

#### 3.3.2. Model Setting

Given that the average level of HQDM for Clubs 1–4 showed a decreasing trend, we constructed ordered discrete variables of clubs on the basis of the identification results of the convergence clubs, with values 1–4 denoting Clubs 1–4, respectively (i.e., Club 1 takes the value of 1, Club 2 takes the value of 2, and so on). In order to identify and analyze the causes of club convergence, we used an ordered logit model to explore the influence of the above variables on the convergence of HQDM clubs. The model was established as follows:(16)$$\mathrm{club}=\beta_{1}ER+\beta_{2}EP+\beta_{3}OPEN+\beta_{4}OMC+\beta_{5}\ln UPD+\varepsilon$$
where club represents the convergence clubs. As the first two clubs each contained only two provinces, referring to Bartkowska and Riedl (2012) [23], the first two clubs were combined into one club as Club 1, and the last two clubs were Clubs 2 and 3, so club = 1, 2 and 3, representing high, medium, and low levels of HQDM, respectively. Furthermore, ε is the perturbation term of the model, obeying a logit distribution with mean 0 and variance $\pi^{2}/3$. As the explanatory variables were cross-sectional data, in order to avoid statistical errors caused by the selection of data for only one year, referring to Bai C. et al., (2020) [24], the values for each explanatory variable were taken as the mean of the values over the period 2008–2019.

#### 3.3.3. Empirical Results

The ordered logit model regression results indicated that the pseudo R^2^ was 0.7155 and the Wald statistic was 16.48, corresponding to a *p*-value of 0.0056. The joint significance of all coefficients in the overall regression model was high, demonstrating that the ordered logit regression model was valid. As ordered logit regression is a nonlinear model using maximum likelihood estimation, its estimated coefficients only reflect the direction of the influence of the explanatory variables on the club division, and do not include marginal effects; therefore, in order to explore the influence of individual variables on the probability of membership in a particular club, we further calculated the marginal effects of each explanatory variable at the mean; the specific results are shown in Table 7.

## 4. Discussion

In this paper, we analyzed the factors influencing the convergence of HQDM clubs, taking 30 provinces of China as the research subject. The regression results in Table 7 demonstrate that environmental regulation (ER) had a negative coefficient at the 1% significance level and, in terms of marginal effects, for every 1 unit increase in environmental regulation at the mean, the probability of belonging to Club 1 or 2 increases by 0.33% and 0.71%, respectively, while the probability of belonging to Club 3 decreases by 1.05%. This suggests that the higher the intensity of environmental regulations, the higher the probability that a region belongs to the high-level HQDM club, consistent with the “innovation compensation effect” proposed by Porter (1991), based on the Porter hypothesis [36]. In other words, environmental regulation can improve the quality of the environment while promoting economic performance, thus achieving a win–win situation in terms of both economic and environmental performance. Environmental regulation can stimulate firms to develop new technologies, reduce costs by reducing resource inputs or increasing efficiency, optimize the quality of production and upgrade the production rating, produce new and more environmentally friendly products [37], and increase firm productivity [38]. Appropriate environmental regulation can promote manufacturing [39,40], which has a significant positive effect on HQDM.

Environmental preference (EP) had a negative coefficient at the 10% level of significance, and the marginal effects indicated that, for every 1-unit increase in environmental regulation at the mean, the probability of belonging to Clubs 1 or 2 increases by 0.55% and 1.18%, respectively, while the probability of belonging to Club 3 decreases by 1.73%. This indicates that the higher the degree of environmental preference, the greater the probability that the region belongs to the high-level club of HQDM. Environmental preference reflects the social environment of green development, and regions with higher environmental preference pay more attention to ecological and environmental protection, insist on achieving a win–win situation between industrial development and environmental protection, and realize green development of the manufacturing industry, which will undoubtedly drive HQDM positively. As such, environmental preference has a significant positive influence on HQDM.

Openness (OPEN) had a negative coefficient at the 5% significance level, and, for every 1 unit increase in the level of openness at the mean, the probability of belonging to Clubs 1 or 2 increases by 0.42% and 0.91%, respectively, while the probability of belonging to Club 3 decreases by 1.33%. This indicates that the higher the level of external openness, the higher the probability that the region belongs to the HQDM high-level club. Opening up to the outside world can increase sensitivity to foreign external shocks; promote technology spillover effects [41]; open up more resources and markets; allow for the absorbing of capital, advanced knowledge, and new technologies from all over the world; generate collision and integration of various experiences and knowledge [28], and stimulate local technological progress and industrial upgrading, thus promoting HQDM. Collectively, the level of openness has a significant positive effect on HQDM.

The coefficient of organizational management change (OMC) was negative at 1% level of significance, and for every 1 unit increase in organizational management change at the mean, the probability of belonging to Clubs 1 or 2 increases by 1.13% and 2.41%, respectively, while the probability of belonging to Club 3 decreases by 3.54%. This indicates that the greater the intensity of organizational management change, the greater the probability that the region belongs to the HQDM high-level club. The strength of organizational management change is reflected by the share of non-state-owned enterprises. Private enterprises are an important microsupport for the efficiency of the Chinese economy, being more innovative and cost-effective [42], having a positive impact on productivity [43], operational management that tends to be more efficient than that of state-owned enterprises [44], scientific and rational changes that are beneficial to enhance marketability, and improved input–output efficiency. All of these aspects promote the level of HQDM, and, so, organizational management changes have a significant positive impact on HQDM.

Urban population density (UPD) had a negative coefficient at the 5% significance level, and for every 1% increase in urban population density at the mean, the probability of belonging to Clubs 1 or 2 increases by 10.26% and 21.99%, respectively, while the probability of belonging to Club 3 decreases by 32.25%. This indicates that the higher the population density, the higher the probability that the region belongs to the HQDM high-level club. Increased population density causes the labor supply and market potential to increase, which enables the development base and sufficient production capacity for manufacturing enterprises and promotes diversification in the manufacturing industry. At the same time, the market potential and consumer demand stimulate enterprises to optimize production and improve industrial development, and, so, the urban population density has a significant positive effect on HQDM.

In summary, environmental regulation and other variables are important factors influencing the convergence of HQDM clubs, all of which presented a positive promoting effect. Regions with greater intensity of environmental regulation, environmental preference, level of openness to the outside world, strength of organizational and management change, and urban population density are more inclined to converge to clubs with higher levels of HQDM, and are less likely to belong to low-level clubs; in other words, enhancement of these variables is conducive to promoting HQDM and reducing regional disparities. Among them, the effect of environmental preference on Club 1 was not significant, while the rest of the variables had significant effects on each club, indicating that environmental preference does not play a significant role in Club 1.

## 5. Conclusions, Recommendations, and Outlook

### 5.1. Conclusions

In this paper, we investigated the spatial and temporal evolution characteristics and club effects of HQDM in China by measuring the level of HQDM in 30 provinces. It was found that: (1) The overall HQDM has gone through the three stages of expansion, cultivation, and promotion, and has shifted from high-growth to high-quality development. (2) There are no manufacturing convergence characteristics in the overall nation or the three traditional economic zones. Notably, the four convergence clubs of HQDM were not significantly similar in the geographical distribution. (3) Factors such as environmental regulation intensity, environmental preference, and so on can significantly affect the category of HQDM convergence club to which a region belongs. The higher the level of these factors, the higher the probability that the region belongs to a high-level club.

### 5.2. Recommendations

In response to the above findings, the following recommendations are made:

First, to promote the development of the manufacturing industry “from big to strong,” we must place the issue of development quality in a more prominent position. Relevant departments should select feasible new development strategies with a new systematic way of thinking and a new concept, and start from the five dimensions of HQDM to form a new power mechanism.

Secondly, according to the divided clubs, relevant departments should adapt to local conditions, formulate differentiated development strategies, and constantly consolidate and optimize the manufacturing development of Clubs 1 and 2, focus on tapping the development potential of Club 3, and inject new development momentum into Club 4. To achieve this, they must give full play to the leading role of the “outstanding and outliers,” pull the progress of the “backward and outliers,” and form a “trickle-down effect,” thus achieving complementary advantages and win–win cooperation.

Third, from the various influencing factors of HQDM club convergence, (1) the government should introduce reasonable policies and regulations, appropriately increase the intensity of environmental regulations, and maximize the incentive effect of environmental regulations on HQDM; (2) while developing manufacturing industries, the greening coverage of cities should be increased to achieve a win–win situation between industrial development and ecological environment optimization; (3) advanced enterprises should strengthen their foreign communication and exchange, absorb advanced knowledge and technology, and optimize their own industrial development; (4) the reform of state-owned enterprises should be actively promoted, particularly in terms of enhancing the social responsibility of state-owned enterprises in independent innovation. Policy protection should be implemented for private enterprises, giving full play to their innovation and efficiency advantages, thus improving the market competition system and developing a reasonable competition mechanism; (5) the efficiency of labor resource allocation should be improved through financial support and the establishment of a sound mechanism for releasing information on labor supply and demand, among other aspects, in order to realize the transformation from quantitative to qualitative advantages of the labor force. 

### 5.3. Outlook

Overall, this study revealed convergent patterns and regional differences in HQDM. Future work may consider differences specific to the prefecture level or enterprise level, in order to analyze the club effect of high-quality manufacturing development from a more microscopic perspective. In addition, the development paths of different clubs can be explored, according to the various dimensions of HQDM.

## Figures and Tables

**Figure 1 ijerph-19-16228-f001:**
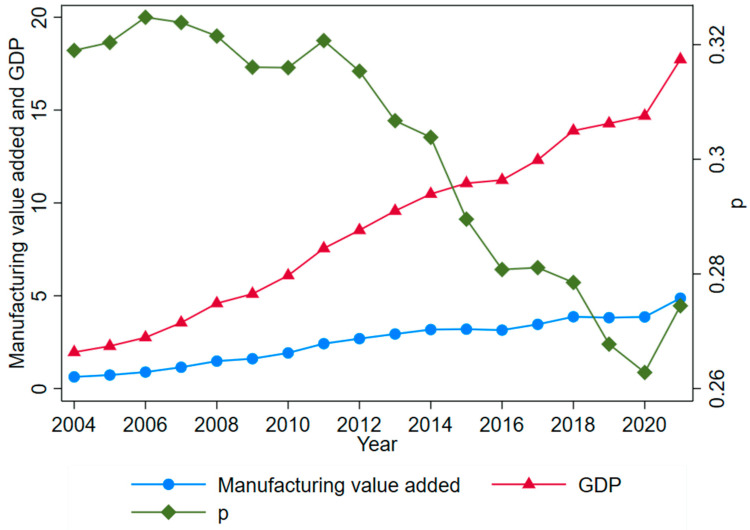
Trends of manufacturing value added and GDP in China. Notes: Data source for Figure 1 is the World Bank; unit: trillions of dollars (current dollar); p, the share of manufacturing value added in GDP.

**Figure 2 ijerph-19-16228-f002:**
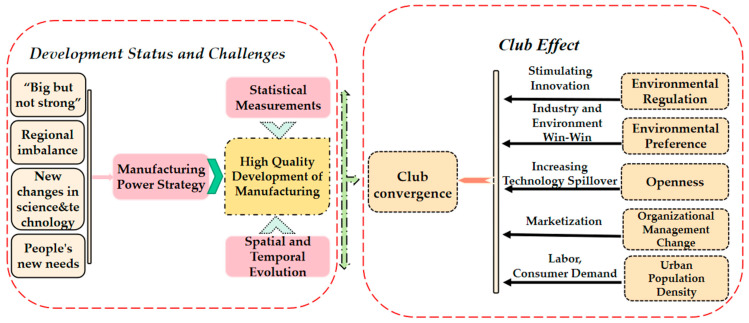
Mechanistic transmission diagram of the statistical measures and club effects of HQDM.

**Figure 3 ijerph-19-16228-f003:**
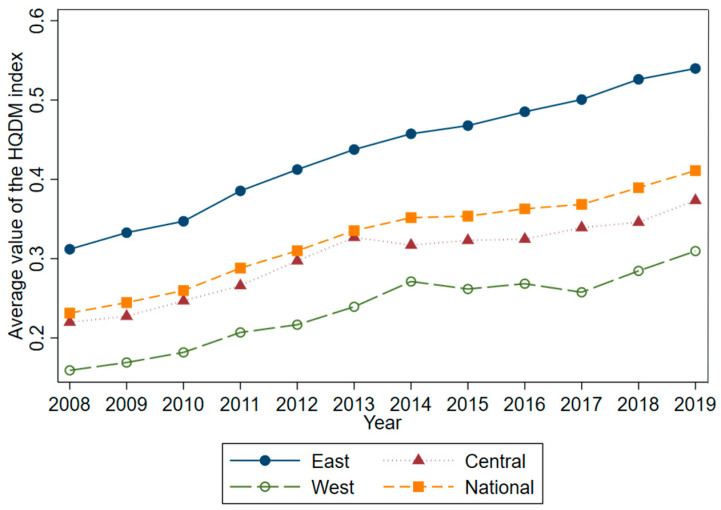
Change in the average value of the HQDM index, both nationally and regionally, for the period 2008–2019.

**Figure 4 ijerph-19-16228-f004:**
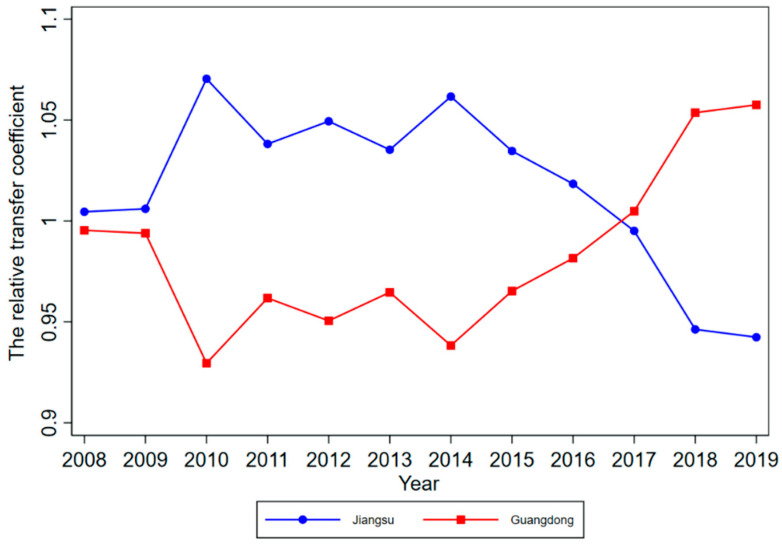
Relative transfer paths of HQDM levels in Club 1 member provinces.

**Figure 5 ijerph-19-16228-f005:**
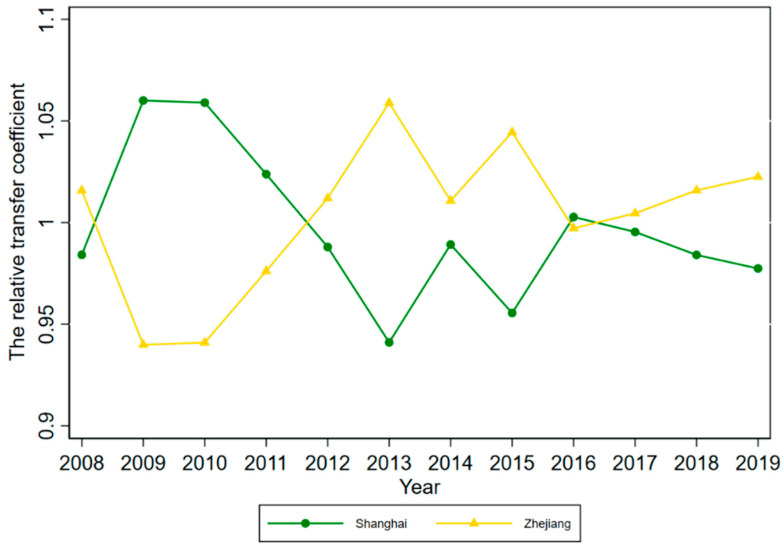
Relative transfer paths of HQDM levels in Club 2 member provinces.

**Figure 6 ijerph-19-16228-f006:**
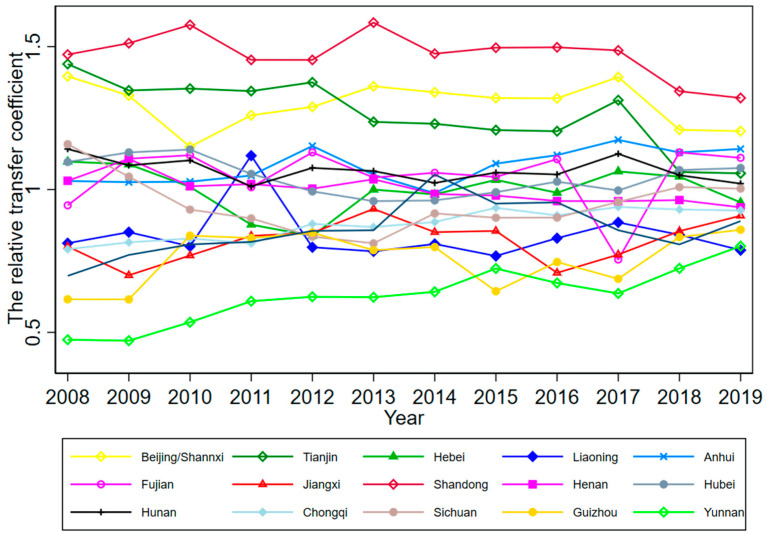
Relative transfer paths of HQDM levels in Club 3 member provinces.

**Figure 7 ijerph-19-16228-f007:**
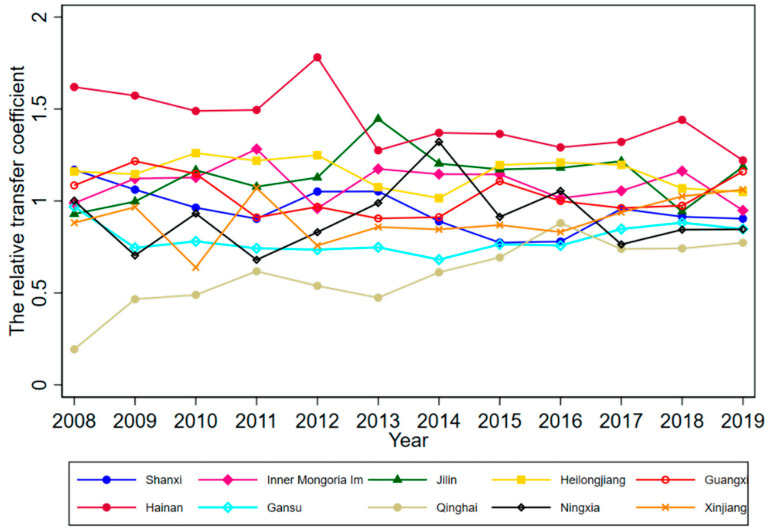
Relative transfer paths of HQDM levels in Club 4 member provinces.

**Table 1 ijerph-19-16228-t001:** HQDM comprehensive evaluation index system and index weights.

Tier 1 Indicators	Secondary Indicators	Indicator Description	Properties	Variable Name
Innovation-driven	R&D input	R&D expenditure	+	*X* _1_
	New product development input	Expenses for new product development	+	*X* _2_
	Number of R&D institutions	Number of R&D institutions opened	+	*X* _3_
	R&D institutional input	Expenses for setting up R&D institutions	+	*X* _4_
	R&D output	Number of R&D projects	+	*X* _5_
	New product project output	Number of new product development projects	+	*X* _6_
	Benefits of new products	Revenue from sales of new products	+	*X* _7_
	Patent output	Number of valid invention patents	+	*X* _8_
	Technical Contribution	Contract turnover of technology market	+	*X* _9_
Quality-first	Total labor productivity	Industrial value added/Average number of all employees	+	*X* _10_
	Technology introduction	Expenditure on digestion and absorption of introduced technologies	+	*X* _11_
	Equipment transformation	Expenditures for technological transformation	+	*X* _12_
	Brand value level	Brand value/Industrial value added	+	*X* _13_
	Product quality level	Superiority rate	+	*X* _14_
	Product Sales	Product sales rate	+	*X* _15_
Green development	Comprehensive utilization rate of industrial solid waste	Comprehensive utilization of solid waste/Generation of solid waste	+	*X* _16_
	Investment intensity of pollution control	The amount of investment completed in industrial pollution control/Industrial value added	+	*X* _17_
	Wastewater treatment	Industrial wastewater treatment facilities treatment capacity	+	*X* _18_
	Energy consumption per unit of industrial value added	Total energy consumption/Industrial value added	-	*X* _19_
Structural optimization	Intelligent manufacturing	Number of broadband Internet access ports	+	*X* _20_
	Level of informatization	Mobile phone subscribers	+	*X* _21_
	Development of new products in high-end industries	Number of new product development projects in high-tech industry/Number of industrial new product development projects%	+	*X* _22_
	Intensity of technological transformation of high-end industries	Expenditure for technological transformation of high-tech industries/Expenditure for technical transformation%	+	*X* _23_
Talent-based	Talent input intensity	Full-time equivalent R&D personnel/Number of Employees	+	*X* _24_
	Treatment of talent	Average wage of manufacturing workers	+	*X* _25_
	Talent intelligence level	Number of doctorates and masters in R&D institutions run by enterprises	+	*X* _26_

Notes: A ”+” indicates that the indicator is a positive indicator; a ”-” indicates that the indicator is a negative indicator.

**Table 2 ijerph-19-16228-t002:** Annual measurement results of HQDM total index by province.

Province	2008	2009	2010	2011	2012	2013	2014	2015	2016	2017	2018	2019	Average Annual Growth Rate (%)
Beijing	0.331	0.326	0.306	0.369	0.398	0.465	0.481	0.477	0.484	0.522	0.488	0.508	3.967
Tianjin	0.341	0.330	0.360	0.393	0.425	0.423	0.441	0.436	0.442	0.492	0.428	0.445	2.456
Hebei	0.260	0.267	0.269	0.257	0.259	0.342	0.353	0.373	0.363	0.398	0.421	0.402	4.048
Shanxi	0.198	0.193	0.179	0.190	0.249	0.262	0.233	0.194	0.199	0.236	0.227	0.243	1.898
Inner Mongolia	0.167	0.204	0.210	0.270	0.227	0.292	0.299	0.287	0.260	0.260	0.288	0.255	3.933
Liaoning	0.192	0.208	0.213	0.327	0.247	0.268	0.290	0.277	0.305	0.332	0.339	0.332	5.085
Jilin	0.157	0.181	0.217	0.227	0.267	0.360	0.315	0.294	0.302	0.299	0.234	0.319	6.652
Heilongjiang	0.196	0.208	0.234	0.256	0.296	0.268	0.265	0.300	0.309	0.295	0.265	0.282	3.364
Shanghai	0.323	0.388	0.409	0.424	0.434	0.424	0.486	0.493	0.548	0.560	0.577	0.614	6.034
Jiangsu	0.404	0.437	0.483	0.535	0.582	0.621	0.646	0.667	0.695	0.733	0.749	0.773	6.071
Zhejiang	0.333	0.344	0.364	0.405	0.444	0.477	0.496	0.539	0.545	0.565	0.596	0.643	6.164
Anhui	0.244	0.251	0.274	0.307	0.356	0.359	0.354	0.394	0.411	0.440	0.456	0.481	6.370
Fujian	0.224	0.271	0.298	0.295	0.349	0.356	0.380	0.377	0.406	0.283	0.456	0.468	6.938
Jiangxi	0.190	0.171	0.205	0.245	0.262	0.319	0.305	0.309	0.260	0.289	0.345	0.383	6.587
Shandong	0.349	0.370	0.420	0.425	0.449	0.542	0.529	0.540	0.550	0.557	0.542	0.556	4.336
Henan	0.244	0.271	0.269	0.298	0.310	0.354	0.353	0.354	0.352	0.359	0.388	0.395	4.475
Hubei	0.260	0.277	0.304	0.309	0.307	0.328	0.345	0.358	0.377	0.373	0.431	0.453	5.188
Hunan	0.271	0.266	0.293	0.296	0.332	0.364	0.367	0.383	0.386	0.422	0.423	0.430	4.306
Guangdong	0.400	0.432	0.419	0.496	0.527	0.578	0.571	0.623	0.670	0.740	0.835	0.867	7.278
Guangxi	0.183	0.221	0.213	0.192	0.229	0.225	0.238	0.278	0.256	0.237	0.241	0.312	4.956
Hainan	0.274	0.286	0.277	0.315	0.422	0.318	0.358	0.342	0.330	0.325	0.357	0.328	1.666
Chongqing	0.187	0.200	0.221	0.237	0.272	0.297	0.318	0.338	0.334	0.352	0.375	0.391	6.904
Sichuan	0.275	0.256	0.248	0.263	0.258	0.278	0.329	0.326	0.331	0.358	0.407	0.423	4.004
Guizhou	0.146	0.151	0.223	0.243	0.262	0.269	0.286	0.233	0.274	0.258	0.336	0.362	8.616
Yunnan	0.112	0.115	0.143	0.178	0.193	0.213	0.230	0.261	0.247	0.238	0.292	0.338	10.524
Shaanxi	0.165	0.189	0.215	0.239	0.264	0.293	0.378	0.343	0.351	0.321	0.326	0.375	7.726
Gansu	0.164	0.135	0.145	0.156	0.174	0.186	0.178	0.192	0.194	0.209	0.219	0.228	3.050
Qinghai	0.033	0.085	0.091	0.130	0.128	0.118	0.160	0.174	0.225	0.182	0.184	0.208	18.312
Ningxia	0.169	0.128	0.173	0.143	0.197	0.246	0.345	0.229	0.270	0.188	0.209	0.228	2.724
Xinjiang	0.149	0.176	0.119	0.225	0.179	0.214	0.221	0.218	0.213	0.231	0.254	0.286	6.097

**Table 3 ijerph-19-16228-t003:** Results of convergence tests for the national and regional club areas.

Test Subjects	Number of Provinces	$\hat{b}$	Standard Error (sd)	t	Convergence or Not
National	30	−1.3514	0.1219	−11.0874	No
East	11	−1.6498	0.1378	−11.9733	No
Central	8	−2.7675	0.2483	−11.1466	No
West	11	−1.1212	0.2361	−4.7495	No

**Table 4 ijerph-19-16228-t004:** Results of convergence test for Chinese provincial clubs.

Initial Clubs	Estimated Value	Merging Test			Final Clubs
Club A	0.2543 (0.1225)	Club A + B −1.2238 (−4.2839)			Club1:Club A
Club B	0.6019 (0.2771)		Club B + C −0.7123 (−2.9357)		Club2:Club B
Club C	−0.1923 (−0.5220)			Club C + D −0.8800 (−6.3737)	Club3:Club C
Club D	0.6068 (4.087)				Club4:Club D

Notes: The values in round brackets are *t*-statistics.

**Table 5 ijerph-19-16228-t005:** Basic characteristics of convergence clubs.

Club	Average Value of HQDM Index	Characteristics	Membership
1	0.6035	High level	Guangdong, Jiangsu.
2	0.4763	Relatively high level	Shanghai, Zhejiang.
3	0.3315	Moderate level	Anhui, Beijing, Chongqing, Fujian, Guizhou, Hebei, Henan, Hubei, Hunan, Jiangxi, Liaoning, Shandong, Shaanxi, Sichuan, Tianjin, Yunnan.
4	0.2303	Low level	Gansu, Guangxi, Hainan, Heilongjiang, Jilin, Inner Mongolia, Ningxia, Qinghai, Shanxi, Xinjiang.

**Table 6 ijerph-19-16228-t006:** Descriptive statistics of variables.

Variable	Obs	Mean	SD	Min	Max
ER	30	65.6269	14.4579	42.8488	113.2422
EP	30	38.7243	3.5536	30.4646	46.8173
OPEN	30	27.0244	28.8921	2.5337	119.0657
OMC	30	89.5992	7.0538	73.7419	98.3274
lnUPD	30	7.8707	0.4039	7.1745	8.5395

**Table 7 ijerph-19-16228-t007:** Regression results of club convergence factors and their marginal effects.

Variables	Coefficient	Marginal Effects		
		Club 1	Club 2	Club 3
ER	−0.1576 *** (−2.60)	0.0033 *** (3.04)	0.0071 *** (2.65)	−0.0105 *** (−3.14)
EP	−0.2607 * (−1.95)	0.0055 (1.61)	0.0118 *** (3.11)	−0.0173 ** (−2.47)
OPEN	−0.1997 ** (−2.36)	0.0042 *** (3.68)	0.0091 *** (2.65)	−0.0133 *** (−3.32)
OMC	−0.5318 *** (−3.50)	0.0113 *** (3.36)	0.0241 *** (3.15)	−0.0354 *** (−3.87)
lnUPD	−4.8485 ** (−2.00)	0.1026 ** (2.18)	0.2199 ** (2.18)	−0.3225 ** (−2.36)

*t*-statistics in parentheses; * *p* < 0.1, ** *p* < 0.05, *** *p* < 0.01.

## Data Availability

The data presented in this study are available on request from the corresponding author.

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
