# Peer review of "Statistical Measurements and Club Effects of High-Quality Development in Chinese Manufacturing"

_ijerph, 2022, doi:10.3390/ijerph192316228_

Round 1

Reviewer 1 Report

The authors have done a fair study and justified the need for the research. However, the following comments need to be addressed:

1. The font size of the mathematical equations seems inconsistent. The equations are not formatted correctly. At may places, the equations are not aligned to text correctly

2. In equation 1, the authors have used Min_Max scaler function for standardization. Explain the reason for the same.

3. What do the positive and negative indicators indicate?

4. In Table 1, on what basis are secondary indicators chosen?

5. Many of the graphs have missing axis titles. Kindly rectify.

6. Figure 4 quality is extremely poor. Authors to check the readability of the figures before submission.

Author Response

Response to Reviewer 1 Comments

Point 1: The font size of the mathematical equations seems inconsistent. The equations are not formatted correctly. At may places, the equations are not aligned to text correctly.

Response 1: Regarding the formatting of mathematical equations mentioned by the reviewer, it has been uniformly revised in the text and correctly aligned with the text, and the font has been standardized. Mathematical equations (1)-(16) have been uniformly numbered.

Point 2: In equation 1, the authors have used Min_Max scaler function for standardization. Explain the reason for the same.

Response 2: Since the units of measurement of the indicators are not uniform, they are standardized before they are used to calculate the composite indicators, i.e., the absolute values of the indicators are converted into relative values.

In equation 1,denotes raw data for indicator  of province  within dimension. is the maximum value of the indicator in all years, is the smallest value of the indicator in all years. We look for the maximum and minimum values of a certain indicator in the panel data, so they are denoted by .

In the entropy method, the positive and negative indicator values represent different meanings (higher values for positive indicators are better, and lower values for negative indicators are better), so for positive and negative indicators, we use different algorithms for data normalization. In order to make the formula expression clearer, with reference to the study of Wang H & Li B in the following literature, equation (1) and (2) have been refined in the paper, and equation (1) is written as:

  ,

Equation (2) is written as:

.

Wang H, Li B. Environmental regulations, capacity utilization, and high-quality development of manufacturing: an analysis based on Chinese provincial panel data. Sci Rep,2021,11:19566.

Point 3: What do the positive and negative indicators indicate?

Response 3: The positive and negative indicators are distinguished by what their values represent. For the target level indicator (HQDM), if a higher value of an indicator is better, then it is a positive indicator; if a lower value of an indicator is better, then it is a negative indicator.

Point 4: In Table 1, on what basis are secondary indicators chosen?

Response 4: Secondary indicators are selected based on the principles of scientificity, representativeness and operability.

Scientificity means that the selection of secondary indicators should follow the law of development of things and make accurate measurement through scientific indicators; representativeness means that secondary indicators can reflect the content of the evaluation of primary indicators and find specific indicators that can quantify the primary indicators through quantitative methods; operability means that the selected indicators are practical and feasible, and the availability of indicator data should be considered, some indicators are suitable but not available Some indicators, though suitable, cannot be obtained, and are not practical and lack operability.

Point 5: Many of the graphs have missing axis titles. Kindly rectify.

Response 5: For the left vertical axis in Figure 1, the axis title has been added: Manufacturing value added and GDP.

For Figure 3, we have added the horizontal axis caption: Year and the vertical axis caption: Average value of the HQDM index have been added.

For Figure 4 (which has been redrawn as Figure 4-7), we have added the horizontal axis caption: Year and the vertical axis caption: The relative transfer coefficient.

Point 6:Figure 4 quality is extremely poor. Authors to check the readability of the figures before submission.

Response 6:As for Figure 4, the size before was too small to make it clear, we redrew the figure, namely Figure 4-7, and unified the size with other figures in the paper

Reviewer 2 Report

I suggest to better explain the relationship between production quality e production rating and environmental effects

Author Response

Response to Reviewer 2 Comments

Point 1: I suggest to better explain the relationship between production quality e production rating and environmental effects.

Response 1: We have considered the comments of the reviewer experts and explained the relationship between environmental regulations and production quality and production levels in our analysis of the impact of the environment. The explanation is as follows:

Environmental regulation can stimulate firms to develop new technologies, reduce costs by reducing resource inputs or increasing efficiency, optimize the quality of production and upgrade the production rating, produce new and more environmentally friendly products, and increase firm productivity.

Reviewer 3 Report

The current revised paper presents a statistical study on the level of high-quality development of manufacturing in China and examines its spatial and temporal evolution. The authors used a nonlinear time-varying factor model for club convergence identification and testing from the data, portray the evolutionary path of each converging club member through the relative transfer path curve, and further examine the determinants of club convergence using an ordered logit model. The authors found that the level of HQDM in China has gone through three stages expansion period, cultivation period, and promotion period.  I have found the paper to be interesting. However, some concerns need to be addressed before accepting the paper for publication to improve the readability and clarity of the manuscript:

-          The article resembles an analytical review of the results of other authors but is not. The main criteria by which the study is conducted have not been disclosed. What are the research objectives? What have the authors done in the presented work?

-          Most of the results are merely described and are limited to comparing the data observation and describing results. The authors are encouraged to include more detailed results and discussion sections and critically discuss the observations from this investigation with existing literature.

Please identify the difference and compare your results with the current studies:

Pan, Wei, Jing Wang, Zhi Lu, Yansui Liu, and Yurui Li. "High-quality development in China: Measurement system, spatial pattern, and improvement paths." Habitat International 118 (2021): 102458.

Fan, C. Cindy, and Allen J. Scott. "Industrial agglomeration and development: a survey of spatial economic issues in East Asia and a statistical analysis of Chinese regions." Economic geography 79, no. 3 (2003): 295-319.

-          In line 435, figure 4 is not clear and data can’t be read.

-          Please consider reviewing the abstract, and highlighting the novelty. I suggest reorganizing the abstract, highlighting the novelties introduced. The abstract should contain answers to some questions, what problem was studied and why is it important? and what conclusions can be drawn from the results? (Please provide specific ones and not generic results, as the abstract should be informative rather than descriptive).

-          The conclusion section is too long. Conclusions should be concise to illustrate the important outputs of the research. Furthermore, where are the recommendations for future work to help other researchers open new windows toward the state of the art in this field.

-          The introduction needs some enhancement, by adding some updated references in 2022. Although we are at the end of 2022, the author only mentioned 2 papers in 2022 which illustrates the oldness of the topic and the researcher's reluctance to work on such a topic.

-          There are a lot of sentences that either does not make sense or lack some additional words to complete their meaning. A thorough review of the article content is needed to improve the quality of the text.

-          The authors should make sure that the format of tables is consistent (i.e. same size, style, format..etc). All figures are different from each other in different sizes and formats.

-          The use of the English language is reasonable, however, there are a number of punctuation and grammatical errors; that should be corrected and rephrased using academic English for a better flow of text for the reader.

-          The authors need to clarify the difference between the traditional artificially predetermined club method and the club convergence identification approach. The accuracy of each approach?

Please, read the text carefully before the next submission of the paper.

Author Response

Response to Reviewer 3 Comments

Point 1: The article resembles an analytical review of the results of other authors but is not. The main criteria by which the study is conducted have not been disclosed. What are the research objectives? What have the authors done in the presented work?

Response 1: The objective of this paper is to examine the spatial and temporal evolutionary characteristics and club effects of high-quality development of manufacturing (HQDM) in China.

Work done by the authors in this paper:

1.In accordance with the principles of scientificity, representativeness and operability, we select indicators that can reflect the HQDM, construct a comprehensive evaluation index system, conduct statistical measurement of the level of HQDM in China, and examine its spatial and temporal evolution characteristics.

2.We introduce a data-driven approach to club convergence identification and testing using a nonlinear time-varying factor model, taking into account individual variability, and find that club convergence effects exist, obtain four converging clubs, and inscribe the evolutionary path of each converging club member through a relative transfer path curve.

3.We examined the causes of the club convergence effect using an ordered logit model to test the convergence mechanism of regional HQDM.

On the whole, we reveal the convergence patterns and regional differences in HQDM, which provides a new perspective for determining the trends of high-quality manufacturing development, thus allowing for policy recommendations targeted at narrowing the manufacturing development gap.

Point 2: Most of the results are merely described and are limited to comparing the data observation and describing results. The authors are encouraged to include more detailed results and discussion sections and critically discuss the observations from this investigation with existing literature.

Please identify the difference and compare your results with the current studies:

Pan, Wei, Jing Wang, Zhi Lu, Yansui Liu, and Yurui Li. "High-quality development in China: Measurement system, spatial pattern, and improvement paths." Habitat International 118 (2021): 102458.

Fan, C. Cindy, and Allen J. Scott. "Industrial agglomeration and development: a survey of spatial economic issues in East Asia and a statistical analysis of Chinese regions." Economic geography 79, no. 3 (2003): 295-319.

Response 2: We carefully read the above 2 references given by the reviewers and reviewed other literature to discuss our results in more detail and compare them with current studies. We mainly revised the HQDM measurement results section of the paper and added the following references.

[1] Wang, M.; Yu, D.; Chen, H.; Li, Y. Comprehensive Measurement, Spatiotemporal Evolution, and Spatial Correlation Analysis of High-Quality Development in the Manufacturing Industry. Sustainability, 2022, 14: 5807.

[2] Wei P , Jing W B , Zhi L C, et al. High-quality development in China: Measurement system, spatial pattern, and improvement paths. Habitat Int, 2021, 118:102458.

[3] Zhou, B.; Wang, N.; Zhang, Z.; Liu, W.; Lu, W.; Xu, R.; Li, L. Research on the Spatial-Temporal Differentiation and Path Analysis of China’s Provincial Regions’ High-Quality Economic Development. Sustainability, 2022, 14: 6348.

[4] Fan C C , Scott A J . Industrial Agglomeration and Development: A Survey of Spatial Economic Issues in East Asia and a Statistical Analysis of Chinese Regions. Econ Geogr, 2003, 79:295-319.

Point 3: In line 435, figure 4 is not clear and data can’t be read.

Response 3: Regarding Figure 4, the previous size is too small resulting in not unclear, we have redrawn the figure, i.e. Figures 4-7, and unified the size with other figures in the paper.

Point 4: Please consider reviewing the abstract, and highlighting the novelty. I suggest reorganizing the abstract, highlighting the novelties introduced. The abstract should contain answers to some questions, what problem was studied and why is it important? and what conclusions can be drawn from the results? (Please provide specific ones and not generic results, as the abstract should be informative rather than descriptive).

Response 4: Following the suggestions made by the reviewers, we reorganized the abstract to highlight the novelty of the paper, the research questions, and the conclusions drawn from the results. The revised abstract is as follows.

Advanced manufacturing is the pillar of building a modern economic system. We measured the level of high-quality development of manufacturing (HQDM) in China, and found that it has gone through the three stages of expansion, cultivation, and promotion. Spatially, it is characterized as "high in the east, low in the west" and "fast in the west, slow in the east", and presents non-equilibrium characteristics. To overcome the subjective bias introduced by artificially set clubs, we utilize a data-driven non-linear time-varying factor model for clustering into four con-vergent clubs, where provinces with higher intensity of environmental regulation and environ-mental preference tend to move closer to the clubs with a higher level of HQDM. We reveal the convergence patterns and regional differences in HQDM, which provides a new perspective for determining the trends of high-quality manufacturing development, thus allowing for policy recommendations targeted at narrowing the manufacturing development gap.

Point 5: The conclusion section is too long. Conclusions should be concise to illustrate the important outputs of the research. Furthermore, where are the recommendations for future work to help other researchers open new windows toward the state of the art in this field.

Response 5: Based on the suggestions made by the reviewers, we have revised our conclusions. We have abridged the original content, briefly described the important results of this study, made relevant policy recommendations, and finally looked forward to future research directions. The revised conclusions are as follows.

5.1 Conclusions

In this paper, we investigated the spatial and temporal evolution characteristics and club effects of HQDM in China by measuring the level of HQDM in 30 provinces. It was found that: (1) The overall HQDM has gone through the three stages of expansion, cultivation, and promotion, and has shifted from high-growth to high-quality development.  (2) There are no manufacturing convergence characteristics in the overall nation or the three traditional economic zones. Notably, the four convergence clubs of HQDM were not significantly similar in the geographical distribution. (3) Factors such as environmental regulation intensity, environmental preference, and so on can significantly affect the category of HQDM convergence club to which a region belongs. The higher the level of these factors, the higher the probability that the region belongs to a high level club.

5.2 Recommendations

In response to the above findings, the following recommendations are made:

First, to promote the development of the manufacturing industry "from big to strong," we must place the issue of development quality in a more prominent position. Relevant departments should select feasible new development strategies with a new systematic way of thinking and a new concept, and start from the five dimensions of HQDM to form a new power mechanism.

Secondly, according to the divided clubs, relevant departments should adapt to local conditions, formulate differentiated development strategies, and constantly consolidate and optimize the manufacturing development of Clubs 1 and 2, focus on tapping the development potential of Club 3, and inject new development momentum into Club 4. To achieve this, they must give full play to the leading role of the "outstanding and outliers," pull the progress of the "backward and outliers," and form a "trickle-down effect," thus achieving complementary advantages and win–win cooperation.

Third, from the various influencing factors of HQDM club convergence, (1) the government should introduce reasonable policies and regulations, appropriately increase the intensity of environmental regulations, and maximize the incentive effect of environmental regulations on HQDM; (2) while developing manufacturing industries, the greening coverage of cities should be increased to achieve a win–win situation between industrial development and ecological environment optimization; (3) advanced enterprises should strengthen their foreign communication and exchange, absorb advanced knowledge and technology, and optimize their own industrial development; (4) the reform of state-owned enterprises should be actively promoted, particularly in terms of enhancing the social responsibility of state-owned enterprises in independent innovation. Implement policy protection for private enterprises, giving full play to their innovation and efficiency advantages, thus improving the market competition system and developing a reasonable competition mechanism; and (5) the efficiency of labor resource allocation should be improved through financial support and the establishment of a sound mechanism for releasing information on labor supply and demand, among other aspects, in order to realize the transformation from quantitative to qualitative advantages of the labor force.

5.3 Outlook

Overall, this study revealed convergent patterns and regional differences in HQDM. Future work may consider differences specific to the prefecture level or enterprise level, in order to analyze the club effect of high-quality manufacturing development from a more microscopic perspective. In addition, the development paths of different clubs can be explored, according to the various dimensions of HQDM.

Point 6: The introduction needs some enhancement, by adding some updated references in 2022. Although we are at the end of 2022, the author only mentioned 2 papers in 2022 which illustrates the oldness of the topic and the researcher's reluctance to work on such a topic.

Response 6: Based on the suggestions of the reviewers, we improved the introduction by adding three new 2022 papers in the second and third paragraphs of the introduction. Among them, the first paper is about the convergence of green total factor productivity in China, the second paper studies the beta convergence of high-tech manufacturing in the EU, and the third paper examines club convergence in R&D expenditure across European regions. By updating the literature, we enriched the research discussion on convergence and enhanced the practical value of this study.

[1] Zhuang, W, Wang, Y, Lu, C-C, Chen, X. The green total factor productivity and convergence in China. Energy Sci Eng. 2022, 10: 2794- 2807.

[2] Erban A C ,  Pelinescu E ,  Dospinescu A S . Beta convergence analysis of gross value added in the high-technology manufacturing industries[J]. Technological and Economic Development of Economy, 2021:1-23.

[3] Kijek T ,  Kijek A ,  Matras-Bolibok A . Club Convergence in R&D Expenditure across European Regions[J]. Sustainability, 2022, 14:832.

Point 7: There are a lot of sentences that either does not make sense or lack some additional words to complete their meaning. A thorough review of the article content is needed to improve the quality of the text.

Response 7: As suggested by the reviewers, we have thoroughly reviewed the content of the article and deleted some meaningless sentences, for example, the abstract and conclusion have been streamlined. In addition, we have improved some incomplete sentences. All contents have been checked and revised by MDPI English Editing.

Point 8: The authors should make sure that the format of tables is consistent (i.e. same size, style, format..etc). All figures are different from each other in different sizes and formats.

Response 8: We have modified the formatting of the tables, including size, style, format...etc., to be consistent. In addition, the lower border of table 1 and the first line of table 6 have been modified to be consistent with other tables. The size and format of all figures (Figure 1-7) have been kept consistent.

Point 9: The use of the English language is reasonable, however, there are a number of punctuation and grammatical errors; that should be corrected and rephrased using academic English for a better flow of text for the reader.

Response 9: Regarding the English language, we have revised the manuscript based on the reviewers' comments and have had it reviewed by MDPI English Editing for punctuation and grammatical errors, and corrected and rephrased using academic English. The English-Editing-Certificate is as follows:

Point 10: The authors need to clarify the difference between the traditional artificially predetermined club method and the club convergence identification approach. The accuracy of each approach?

Response 10: The traditional method of manually booking clubs uses geographic information or city names as a priori information for dividing regions, and then makes convergence judgments on artificially set region boundaries, thus ignoring individual differences and failing to scientifically reflect the heterogeneous characteristics of regional HQDM. Such a division may have sample selection bias and cannot consider the common development trend of different types of regions, which may hide the potential club convergence. The premise of the concept of club convergence is "similar initial level and structural characteristics", but it is difficult to satisfy this point by setting clubs artificially, which will reduce the scientificity of club identification to a certain extent.

The club convergence identification method is a data-driven approach that overcomes the drawbacks of manually set club methods. The logarithmic t-test in this method takes into account regional heterogeneity and is not affected by small sample characteristics and is therefore robust to the smoothness of the series. As far as the clustering algorithm is concerned, Phillips and Sul (2007) argue that it is a data-driven algorithm that can avoid the sample selection errors caused by manual grouping.

Round 2

Reviewer 3 Report

Many thanks for the revision and for incorporating all suggested changes to the manuscript that are nicely reflected. The authors did a good job to improve the article. I believe that article has become much better and now I recommend this article for publication.